# Perception to Adaptation of Climate Change in Nepal: An Empirical Analysis Using Multivariate Probit Model

**Arun GC and Jun-Ho Yeo \***

Department of Agricultural Economics, Kyungpook National University, 80 Daehakro, Daegu 41566, Korea; gcrun88@gmail.com
\* Correspondence: jhyeo@knu.ac.kr

**Abstract:** This study assessed farmers' perception of climate change, and estimated the determinants of, and evaluated the relationship among, adaptation practices using the multivariate probit model. A survey in 300 agricultural households was carried out covering 10 sample districts considering five agro-ecological zones and a vulnerability index. Four adaptation choices (change in planting date, crop variety, crop type and investment in irrigation) were deemed as outcome variables and socioeconomic, demographic, institutional, farm-level and perceptions variables were deployed as explanatory variables. Their marginal effects were determined for three climatic variables—temperature, precipitation and drought. Age, gender and education of head of household, credit access, farm area, rain-fed farming and tenure, were found to be more influential compared to other factors. All four adaptation options were found to be complimentary to each other. Importantly, the intensity of the impact of dependent variables in different models, and for the available adaptation options, were found to be unequal. Therefore, policy options and support facilities should be devised according to climatic variables and adaptation options to achieve superior results.

**Keywords:** adaptation; perception; climate change; Nepal; multivariate probit

## 1. Introduction

Climate change is having an affect worldwide; however, developing, landlocked and island countries, like Nepal, are most vulnerable [1,2] although the contribution of these countries is negligible [3,4]. Notably, climate change has been recognized as the major challenge of the era. Consequently, the Sustainable Development Goals (SDG) incorporated Climate Action as a separate goal (Goal 13), recognizing notions such as its potential to undermine economic gains, its greater impact in developing countries, and the fact that over 100 million people may become poor and hungry by 2030 if urgent action is not taken, which was not included in the Millennium Development Goals (MDG) [5]. Acknowledging the unique cause and effect relationship between agriculture and climate change, agriculture is the prime concern in the study of climate change. Firstly, it contributes one-third of all Greenhouse Gas (GHG) emissions; among this, 86% is anthropogenic [6]. Secondly, agriculture and pasture claim around 1.3 billion hectares and 3.5 billion hectares respectively, and are, therefore, the basis of the livelihoods of a significant amount of the population [7]. Finally, the sector will be responsible for feeding over nine billion people by 2050 [8].

Changes in cropping patterns, variations in crop yields, and increases in the incidence of disease pests have been attributed to climate change [4]. Moreover, climate change and agriculture, mainly from a food security perspective, have been studied by various scholars [9–12]. Internalizing the potential impact of climate change in agriculture, and ultimately on overall development, the SDG

has incorporated both climate change and agriculture with due priority which could contribute to sustainable development.

Climate change adaptation is relatively a novel notion, which is broadly defined as local-level adjustments to tackle changes under given socioeconomic constraints [5,13,14]. Although climate change adaptation literature has grown recently, measuring determinants are still relatively scarce [13]. Among them, the majority of studies are focused on Africa [15–19]. Yegbemey et al. (2013) assessed farmers' decisions to adapt to climate change in maize farming in West Africa. Mulwa et al. (2017) examined the role of household and farm characteristics along with information in Malawi. Nhemachena et al. (2014) analyzed the determinants of adaptation at a farm level in South Africa. Likewise, few studies have been carried out in Asia. Ung et al. (2016) studied climate change adaptation in costal Cambodia [20]. Gentle and Maraseni (2012) studied adaptation practices in a rural mountain community in Nepal [21].

Nepal is among the most vulnerable countries to climate change [22,23], and its vulnerability is heightening owing to several physical and socioeconomic constraints [2,4,24], where climatic change has already been observed and recorded. A trend analysis of the period 1975–2005 found an annual average temperature increase of 0.06 °C, and a mean precipitation decline of 3.7 mm per month per decade [3]. The predicted increase in annual mean temperature by 2060 is 1.3–3.8 °C under various scenarios, whereas annual average precipitation has been projected to decrease by 10% to 20% [3]. Consequently, water shortages, extreme weather and increased incidence of disease-pests are affecting agriculture, and thus, seriously threatening food security [22].

Since Nepal is indexed as highly vulnerable, comprehensive study on climate change is urgent, and agriculture is a critical area to consider [25]. However, previous studies are mostly descriptive [2,26–28], and a few studies which tried to determine factors affecting adaptations were focused on a specific locality [13,29]. Therefore, this study presents a national perspective, and invokes several factors affecting the adaptation choices of farmers which were not considered before.

## 2. Materials and Methods

Nepal has five physiographic regions—High Mountain, Mountain, Hill, Shivalik and Terai [24]. Table 1 presents the sample districts, physiographic regions and vulnerability index from the National Adaptation Programme of Action (NAPA).

**Table 1.** Sample districts.

| District | Physiographic Region | Vulnerability Index |
|---|---|---|
| Mustang | High Mountain | Moderate |
| Kaski | Mountain | Moderate |
| Rasuwa | Mountain | Moderate |
| Dedeldhura | Hill | Moderate |
| Rolpa | Hill | Moderate |
| Dadhing | Hill | High |
| Terhathum | Hill | Low |
| Chitwan | Shivalik | High |
| Bardiya | Tarai | Low |
| Parsa | Tarai | High |

A pilot was carried out in Dadhing, and a household survey was conducted from January 2015 to June 2015, adopting a random sampling technique. Per district, 30 households were picked to represent 300 households. Respondents were asked whether they had perceived a change in the climate, with possible answers being simply yes or no. If they had perceived changes, then they were further asked whether they had adopted given sets of adaptation options, with possible answers of yes or no.

The simplest model for binary responses is the linear probability model, but it faces the problem of disturbance terms which become deterministic and heteroskedastic; furthermore, the predicted probability could be over 1 or below 0 for extreme values of dependent variables [30,31]. To overcome the deterministic problem, a maximum likelihood estimation technique could be deployed, while to overcome heteroskedasticity, either logit or probit estimations could be employed [30]. The next alternative is a univariate model; however, this cannot measure the potential correlation among unobserved disturbances and the relationship between different choices [32]. To overcome this drawback, the multivariate probit model (MVP) could be employed [13,17,32].

The MVP is a binary response regression model used to estimate both the observed and unobserved influence on dependent variables of several independent variables simultaneously, which permits error terms to correlate freely [33–35].

The general specification for the MVP is [31]:

$$y_m^* = x_m'\beta_m + \varepsilon_m, \ y_m = 1 \ if \ y_m^* > 0, \ 0 \ otherwise, \ m = 1, \ldots\ldots, M, \tag{1}$$

$$E[\varepsilon_m | x_1, \ldots, x_M] = 0$$

$$Var[\varepsilon_m | x_1, \ldots, x_M] = 1$$

$$Cov[\varepsilon_j \varepsilon_m | x_1, \ldots, x_M] = \rho_{jm}$$

$$(\varepsilon_1, \ldots\ldots, \varepsilon_M) \sim N_M[0, R]$$

where $x$ is a matrix of covariates consisting of independent variables, $\beta$ is matrix of unknown regression coefficients and $\varepsilon_m$ is residual error. R is the variance-covariance matrix. The off-diagonal elements in correlation matrix $\rho_{jm}$ represent the unobserved correlation between the stochastic component of the jth and mth options [13].

The marginal effects of independent variables on the propensity to adopt different adaptation options were calculated by the following equation [17]:

$$\frac{\partial P_i}{\partial x_i} = \varphi(x'\beta)\beta_i, \ i = 1, 2, 3, \ldots\ldots, n \tag{2}$$

where $P_i$ is the likelihood of event $i$ which increases adoption of each adaptation option, and $\varphi(.)$is the standard univariate normal density distribution function.

For data analysis, STATA 13.0 was employed, and to get consistent estimates, the number of draws was set to 100, which is five by default [13,36].

Factors affecting the choice of adaptation practices were taken as explanatory variables, as described in Table 2, with the expected relationship. Importantly, there is lack of straightforward theory regarding the selection of predictors [17]; however, several previous empirical studies have enabled the nomination of appropriate predictors.

**Table 2.** Explanatory variables.

| Variable | Description | Mean | Std. Dev. | Expected |
|---|---|---|---|---|
| Farm Decision | Decision maker on farming (Head of Household (HoH) = 1; otherwise 0) | 0.830 | 0.376 | ± |
| HHSize | Household size | 6.021 | 3.101 | ± |
| AgeHoH | Age of HoH in year | 50.90 | 12.44 | ± |
| GenderHoH | Gender of HoH (Male = 1; Female = 0) | 0.0951 | 0.294 | ± |
| EducHoH | Education of HoH in years | 7.173 | 4.937 | + |
| Telephone | Access to the telephone (Yes = 1; No = 0) | 0.933 | 0.250 | + |

**Table 2.** *Cont.*

| Variable | Description | Mean | Std. Dev. | Expected |
|---|---|---|---|---|
| MorePlot | Having more than one plot (Yes = 1; No = 0) | 0.703 | 0.458 | ± |
| FarmArea | Farming area in hectares | 1.068 | 2.363 | ± |
| Tenure | Tenure (Own = 1; otherwise 0) | 0.878 | 0.327 | + |
| PctOnfarmIncome | Percentage of farm income to total income | 76.65 | 36.728 | ± |
| BorrowedYN | Access to the credit (Yes = 1; No = 0) | 0.405 | 0.492 | + |
| NumAdultMale | Number of adult males working on the farm | 1.413 | 0.885 | + |
| Membership | Having membership to any kind of cooperative (Yes = 1; No = 0) | 0.683 | 0.466 | + |
| Distance | Distance to nearest market in kilometers | 4.361 | 8.151 | − |
| TempYN | Perceive change in temperature (Yes = 1; No = 0) | 0.920 | 0.272 | ± |
| DrghtYN | Perceive change in drought frequency (Yes = 1; No = 0) | 0.663 | 0.473 | ± |
| PestYN | Perceive change in incidence of disease pests (Yes = 1; No = 0) | 0.893 | 0.309 | ± |
| Precip_DryWet | Direction of rainfall change (Increasing = 1, Decreasing = 0) | 0.190 | 0.393 | ± |

## 3. Results

Climate change perception was assessed among farmers using three climatic variables instead of asking directly whether they were perceiving it. The majority of the population perceived temperature alteration (92%), precipitation change (87.67%) and change in drought frequency (66.76%).

Among the farmers who perceived temperature alteration, a higher proportion (84.12%) reported an increasing trend and 63.88% responded that precipitation is decreasing. Similarly, 81% of farmers confirmed that the frequency of droughts is increasing.

The adoption of adaptation options according to changes in climatic variabilities are presented in Figure 1. The results show that investment in irrigation facilities is the most common adaptation practice to changes in temperature and drought frequency. Likewise, change in planting date is the most common adaptation option to change in precipitation. Change in crop type was found to be the least common against all kinds of climatic variabilities.

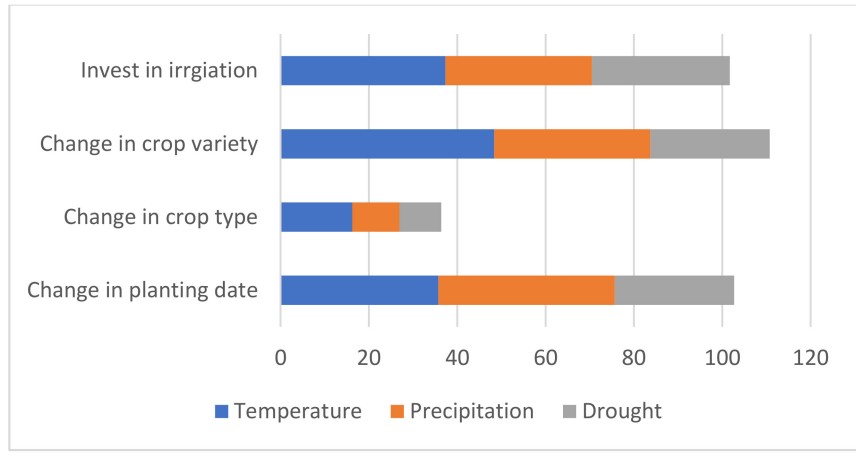

**Figure 1.** Adoption of adaptation options to changes in climatic variables.

The null hypothesis for the test of independence was rejected for all models, as the likelihood ratio test (Prob > $\chi^2$ = 0.0000) of independence of error terms was significant. Thus, the use of MVP is justified, indicating that the model captured wider effects than the single equation-probit model would. Thus, the equations are interdependent. Similarly, all pairwise correlation coefficients (Rho), which are presented in Table 3, are positive and mostly highly significant. This indicates that all four sets of adaptation options are complimentary to each other.

### 3.1. Determinants of Climate Change Adaptation

Table 3 presents the determinants of climate change adaptation for change in temperature, change in precipitation and change in drought frequency, i.e., the temperature model, precipitation model and drought model, respectively. Each model has four dependent variables: change in cropping date (Crop_Date), change in crop type (Crop_Type), change in crop variety (Crop_Vrty) and change in investment on irrigation (Crop_Irrig). It is important to note that each independent variable has a different level of influence on different dependent variables in different models.

In the temperature model, credit access and the percentage of farm income relative to total income were found to be the most influencing independent variables, yielding significant results for three dependent variables. Gender of head of household, membership to any farming institution, farming area and access to weather information were found to be other influencing determinants which produced significant results for at least two dependent variables. Distance to the nearest market not was found to influence any dependent variable, while the remaining factors were found to be significant for at least one. Gender of head of household, household size, many parcels of land and farming decision taken by head of household produced mostly negative implications for adaption in the temperature model, but positive implications elsewhere.

In the precipitation model, perception of more precipitation and gender of head of household were found to be the most influencing determinants, having significant effects on three dependent variables. Tenure, credit access, membership, percentage of farm income, total farming area, having rainfed farming and access to the weather information yielded significant results for at least two dependent variables. Likewise, perceiving change in temperature and drought, farm decision taken by head of household, age of head of household, access to telephone and nearest distance to the market produced significant results for at least one dependent variable. However, the number of adult males working on the farm and perception of change in disease pest prevalence did not yield significant results for any dependent variable. Regarding the direction of influence, credit access, membership, percentage of farm income, farming area, perception of change in drought, age of head of household and telephone access exhibited positive relationships. However, tenure and rainfed farming produced mix results, and the rest yielded negative relationships.

In the drought model, tenure, credit access, number of adult males working on the farm, farming area and rainfed farming were found to be the most influencing determinants, yielding significant results with at least two dependent variables, while percentage of farm income produced no significant results. The remaining factors yielded at least one significant result for dependent variables. Regarding the direction of influence, only number of farm adult males, gender of head of household and distance to the nearest market had negative relationships; the rest had positive relationships.

### 3.2. Marginal Effects

The parameter estimates of the MVP give the likelihood of the occurrence of a given outcome. However, it cannot be quantified. Nevertheless, the marginal effect can be calculated for each model, which gives quantification of the influence of the variables.

Table 3. Results of multivariate probit model (MVP) analysis of determinants of adaptation.

| Models | Temperature Model | | | | Precipitation Model | | | | Drought Model | | | |
|---|---|---|---|---|---|---|---|---|---|---|---|---|
| VARIABLES | Crop_Date | Crop_Type | Crop_Vrty | Crop_Irrig | Crop_Date | Crop_Type | Crop_Vrty | Crop_Irrig | Crop_Date | Crop_Type | Crop_Vrty | Crop_Irrig |
| AgeHoH | 0.015 ** | −0.006 | 0.011 | 0.003 | −0.008 | 0.0133 | 0.0187 ** | −0.006 | 0.0145 | −0.00552 | 0.0228 ** | 0.00121 |
|  | (−0.007) | (−0.009) | (−0.009) | (−0.008) | (−0.007) | (−0.010) | (−0.008) | (−0.007) | (−0.010) | (−0.011) | (−0.010) | (−0.010) |
| GenderHoH | 0.474 * | −0.251 | −0.497 * | −0.138 | 0.19 | −4.827 *** | −1.070 *** | −0.706 ** | −0.0813 | −0.063 | −1.422 ** | −0.406 |
|  | (−0.271) | (−0.375) | (−0.293) | (−0.323) | (−0.277) | (−0.577) | (−0.365) | (−0.350) | (−0.408) | (−0.524) | (−0.586) | (−0.520) |
| Tenure | 0.156 | 0.172 | 0.638 ** | 0.295 | 0.312 | −0.541 * | 0.517 * | −0.285 | 0.539 * | −0.0639 | 0.542 * | −0.171 |
|  | (−0.246) | (−0.290) | (−0.281) | (−0.217) | (−0.253) | (−0.282) | (−0.269) | (−0.266) | (−0.311) | (−0.336) | (−0.327) | (−0.287) |
| BorrowedYN | 0.444 ** | −0.054 | 0.719 *** | 0.445 *** | 0.346 * | 0.281 | 0.3 | 0.360 * | 0.151 | 0.216 | 0.397 * | 0.436 ** |
|  | (−0.174) | (−0.193) | (−0.175) | (−0.172) | (−0.182) | (−0.244) | (−0.187) | (−0.190) | (−0.210) | (−0.261) | (−0.221) | (−0.205) |
| FarmAdultMales | 0.014 | 0.224 * | 0.091 | −0.019 | −0.007 | −0.067 | −0.121 | 0.047 | −0.196 ** | −0.0511 | −0.298 *** | −0.0413 |
|  | (−0.094) | (−0.133) | (−0.105) | (−0.099) | (−0.072) | (−0.083) | (−0.075) | (−0.079) | (−0.096) | (−0.094) | (−0.099) | (−0.095) |
| Membership | 0.005 | 0.294 | 0.481 ** | 0.626 *** | −0.048 | 0.351 | 0.407 * | 0.485 ** | − | − | − | − |
|  | (−0.185) | (−0.231) | (−0.192) | (−0.190) | (−0.191) | (−0.281) | (−0.229) | (−0.216) | − | − | − | − |
| PctOnfarmIncome | −0.002 | 0.009 ** | 0.005 ** | 0.008 *** | −0.001 | 0.005 | 0.009 *** | 0.005 * | −0.002 | −0.001 | −0.003 | 0.005 |
|  | (−0.003) | (−0.004) | (−0.002) | (−0.003) | (−0.003) | (−0.004) | (−0.003) | (−0.003) | (−0.003) | (−0.004) | (−0.003) | (−0.003) |
| Distance | 0.005 | 0.002 | −0.009 | −0.01 | −0.011 | −0.012 | −0.042 ** | −0.019 | 0.00226 | −0.0165 | −0.0294 | −0.0290 * |
|  | (−0.010) | (−0.010) | (−0.008) | (−0.009) | (−0.008) | (−0.015) | (−0.019) | (−0.014) | (−0.013) | (−0.028) | (−0.018) | (−0.016) |
| FarmArea | 0.255 ** | 0.178 | 0.194 * | 0.159 | 0.351 *** | 0.024 | 0.242 ** | 0.079 | 0.464 *** | 0.0272 | 0.290 * | 0.148 |
|  | (−0.123) | (−0.112) | (−0.110) | (−0.103) | (−0.134) | (−0.154) | (−0.105) | (−0.116) | (−0.129) | (−0.153) | (−0.151) | (−0.146) |
| EducHoH | 0.016 | −0.01 | 0.039 * | 0.015 | 0.005 | −0.005 | 0.003 | 0.013 | 0.007 | 0.043 * | 0.0003 | 0.006 |
|  | (−0.017) | (−0.018) | (−0.022) | (−0.019) | (−0.017) | (−0.016) | (−0.019) | (−0.016) | (−0.018) | (−0.023) | (−0.018) | (−0.019) |
| iRainfed | 0.435 ** | −0.116 | −0.028 | 0.254 | 0.112 | −0.447 * | 0.142 | 0.362 * | 0.227 | −0.0758 | 0.442 * | 0.926 *** |
|  | (−0.179) | (−0.200) | (−0.179) | (−0.182) | (−0.183) | (−0.231) | (−0.190) | (−0.200) | (−0.238) | (−0.265) | (−0.235) | (−0.239) |
| WeatherInfo | −0.220 | −0.329 * | −0.458 ** | −0.297 | −0.291 * | −0.521 ** | −0.112 | −0.089 | − | − | − | − |
|  | (−0.177) | (−0.198) | (−0.180) | (−0.183) | (−0.175) | (−0.232) | (−0.193) | (−0.185) | − | − | − | − |
| HHSize | −0.015 | −0.149 ** | 0.013 | −0.004 | − | − | − | − | − | − | − | − |
|  | (−0.040) | (−0.059) | (−0.042) | (−0.041) | − | − | − | − | − | − | − | − |
| MorePlot | −0.041 | 0.04 | −0.144 | −0.317 * | − | − | − | − | − | − | − | − |
|  | (−0.190) | (−0.211) | (−0.191) | (−0.187) | − | − | − | − | − | − | − | − |
| Telephone | − | − | − | − | 0.544 | −0.168 | 0.754 ** | 0.601 | 0.854 * | 0.139 | 0.555 | 0.295 |
|  | − | − | − | − | (−0.354) | (−0.451) | (−0.325) | (−0.394) | (−0.469) | (−0.527) | (−0.396) | (−0.416) |
| TempYN | − | − | − | − | 0.129 | 0.167 | −0.767 ** | −0.45 | − | − | − | − |
|  | − | − | − | − | (−0.335) | (−0.382) | (−0.311) | (−0.305) | − | − | − | − |
| DrghtYN | − | − | − | − | 0.032 | 0.148 | 0.619 ** | 0.297 | − | − | − | − |
|  | − | − | − | − | (−0.198) | (−0.263) | (−0.250) | (−0.224) | − | − | − | − |
| PestYN | − | − | − | − | (0.004) | (0.000) | (−0.022) | (−0.150) | − | − | − | − |
|  | − | − | − | − | (−0.294) | (−0.384) | (−0.333) | (−0.329) | − | − | − | − |
| Precip_DryWet | − | − | − | − | −0.454 ** | −0.014 | −0.573 *** | −0.479 ** | − | − | − | − |
|  | − | − | − | − | (−0.184) | (−0.223) | (−0.215) | (−0.209) | − | − | − | − |
| FarmDecisions | − | − | − | − | 0.166 | −0.232 | −0.151 | −0.572 ** | − | − | − | − |
|  | − | − | − | − | (−0.249) | (−0.309) | (−0.273) | (−0.256) | − | − | − | − |

**Table 3.** *Cont.*

| Models | Temperature Model | | | | Precipitation Model | | | | Drought Model | | | |
|---|---|---|---|---|---|---|---|---|---|---|---|---|
| VARIABLES | Crop_Date | Crop_Type | Crop_Vrty | Crop_Irrig | Crop_Date | Crop_Type | Crop_Vrty | Crop_Irrig | Crop_Date | Crop_Type | Crop_Vrty | Crop_Irrig |
| Constant | −1.712 *** | −1.213 * | −2.364 *** | −1.844 *** | −1.015 | −1.405 | −2.803 *** | −0.69 | −2.839 *** | −1.284 | −2.472 *** | −1.859 ** |
| | (−0.533) | (−0.684) | (−0.618) | (−0.575) | (−0.823) | (−0.936) | (−0.881) | (−0.842) | (−0.796) | (−0.896) | (−0.748) | (−0.789) |
| | Rho1 | Rho2 | Rho3 | | Rho1 | Rho2 | Rho3 | | Rho1 | Rho2 | Rho3 | |
| Rho2 | 0.373 *** | | | | 0.015 | | | | 0.163 | | | |
| Rho3 | 0.408 *** | 0.524 *** | | | 0.274 ** | 0.661 *** | | | 0.597 *** | 0.426 ** | | |
| Rho4 | 0.308 *** | 0.309 *** | 0.956 *** | | 0.065 | 0.620 *** | 0.856 *** | | 0.275 ** | 0.225 | 0.896 *** | |
| Observations | 261 | | | | 251 | | | | 189 | | | |
| Draws | 100 | | | | 100 | | | | 100 | | | |
| Log pseudo likelihood | −515.304 | | | | −449.23 | | | | −328.835 | | | |
| Wald chi2(56) | 182.73 | | | | 1120.69 | | | | 112.09 | | | |
| Prob > chi2 | 0.000 | | | | 0.000 | | | | 0.000 | | | |

| | | |
|---|---|---|
| Likelihood ratio test of rho21 = rho31 = rho41 = rho32 = rho42 = rho43 = 0: chi2(6) = 87.9323 Prob > chi2 = 0.000 Robust standard errors in parentheses *** $p < 0.01$, ** $p < 0.05$, * $p < 0.1$ | Likelihood ratio test of rho21 = rho31 = rho41 = rho32 = rho42 = rho43 = 0: chi2(6) = 72.666 Prob > chi2 = 0.000 | Likelihood ratio test of rho21 = rho31 = rho41 = rho32 = rho42 = rho43 = 0: chi2 (6) = 56.6392 Prob > chi2 = 0.0000 Robust standard errors in parentheses; *** $p < 0.01$, ** $p < 0.05$, * $p < 0.1$ |

Figures in parentheses are standard error.

### 3.2.1. Marginal Effect of the Temperature Change Model

Table 4 portrays the marginal effect of explanatory variables for the temperature model. Age, gender and education of head of household (HoH), area, tenure, credit access, membership, distance, irrigation and number of males increased the probability of adaptation, whereas percentage of farm income, household size, weather information and additional plots decreased it. The largest effect was found for gender, followed by credit access and irrigation. Furthermore, with HoH being a female, the probability of adaptation increased by 0.474. Similarly, access to credit increased the probability of adoption adaptations by 0.445; for the rain-fed farming system, it was 0.435. Similarly, the highest decline was found for having access to weather information (0.22).

**Table 4.** Marginal effect after multivariate probit analysis due to change in temperature.

| Variable | dy/dx | Std. Err. | z | P > z | 95% | C.I. | X |
|---|---|---|---|---|---|---|---|
| **y = Linear Prediction (Predict) = −0.38318332** | | | | | | | |
| AgeHoH | 0.015 | 0.007 | 2.14 | 0.033 | 0.001 | 0.028 | 50.536 |
| GenderHoH * | 0.474 | 0.271 | 1.75 | 0.081 | −0.058 | 1.005 | 0.01 |
| EducHoH | 0.016 | 0.017 | 0.95 | 0.343 | −0.017 | 0.048 | 7.215 |
| FarmArea | 0.255 | 0.123 | 2.08 | 0.037 | 0.015 | 0.496 | 0.933 |
| Tenure * | 0.156 | 0.246 | 0.64 | 0.525 | −0.325 | 0.637 | 0.866 |
| PctOnfarmIncome | −0.002 | 0.003 | −0.94 | 0.35 | −0.008 | 0.003 | 70.876 |
| BorrowYN * | 0.445 | 0.174 | 2.56 | 0.011 | 0.104 | 0.785 | 0.399 |
| HHSize | −0.016 | 0.04 | −0.38 | 0.702 | −0.095 | 0.064 | 5.927 |
| Membership * | 0.005 | 0.185 | 0.02 | 0.98 | −0.359 | 0.368 | 0.69 |
| Distance | 0.005 | 0.01 | 0.44 | 0.661 | −0.016 | 0.025 | 4.308 |
| Weatherinfo * | −0.220 | 0.177 | −1.24 | 0.214 | −0.567 | 0.127 | 0.667 |
| iRainfed * | 0.435 | 0.18 | 2.42 | 0.015 | 0.083 | 0.786 | 0.628 |
| MorePlot * | −0.041 | 0.19 | −0.22 | 0.829 | −0.413 | 0.331 | 0.694 |
| FarmAdultMales | 0.014 | 0.094 | 0.15 | 0.881 | −0.170 | 0.198 | 2.157 |

(*) dy/dx is for discrete change of dummy variable from 0 to 1.

### 3.2.2. Marginal Effect for Precipitation Model

The result of the marginal effect for the precipitation model is presented in Table 5. Perception of temperature and drought frequency changes, farming decision by HoH, female as HoH, own-land, credit and telephone access, farm area, education of HoH and rain-fed farming system increased the probability of adaptation against precipitation; in contrast, higher precipitation, age of HoH, number of adult males, membership, percentage of farm income and distance decreased it. Moreover, possessing a telephone boosted the probability of adaptation by 0.544, which was the highest influence among the explanatory variables, followed by farming area, access to credit and tenure. Similarly, increase in each hectare of land increased the probability of adaptation by 0.351, and if the household had credit access and had its own farming land, the probability of adaptation increased by 0.346 and 0.312 respectively. The highest negative influencing factor was increasing rainfall (0.454).

### 3.2.3. Marginal Effect for the Drought Model

The result of the marginal effect for the drought model is presented in Table 6. Except gender of HoH, adult male and percentage of farm income, all variables imparted a positive influence. Possession of a telephone was quite an influencing factor, increasing the probability of adaptation by 0.854, followed by tenure and farming area. Furthermore, if a household had its own farming land, the probability of adaptation increased by 0.539. Similarly, for every additional hectare of land, the probability of adoption increased by 0.464. In the case of negative influence, the number of male adults was the highest. Increasing male adults by one decreased the probability of adoption by 0.200.

**Table 5.** Marginal effect after multivariate probit analysis due to change in precipitation.

| Variable | dy/dx | Std. Err. | Z | P > z | 95% | C.I. | X |
|---|---|---|---|---|---|---|---|
| **y = Linear Prediction (Predict) = −0.265** | | | | | | | |
| TempYN * | 0.129 | 0.335 | 0.39 | 0.699 | −0.525 | 0.785 | 0.928 |
| DrghtYN * | 0.032 | 0.198 | 0.16 | 0.872 | −0.357 | 0.421 | 0.725 |
| PestYN * | 0.004 | 0.294 | 0.01 | 0.989 | −0.571 | 0.58 | 0.896 |
| Precipitation * | −0.454 | 0.184 | −2.47 | 0.013 | −0.814 | −0.094 | 0.367 |
| FarmDecison * | 0.166 | 0.249 | 0.67 | 0.506 | −0.323 | 0.654 | 0.849 |
| AgeHoH | −0.008 | 0.007 | −1.06 | 0.288 | −0.022 | 0.007 | 51.147 |
| GenderHoH * | 0.19 | 0.278 | 0.68 | 0.495 | −0.354 | 0.733 | 0.096 |
| Tenure * | 0.312 | 0.253 | 1.23 | 0.218 | −0.184 | 0.807 | 0.861 |
| BorrowedYN * | 0.346 | 0.182 | 1.9 | 0.058 | −0.011 | 0.702 | 0.43 |
| NumAdultMale | −0.007 | 0.072 | −0.1 | 0.918 | −0.148 | 0.133 | 2.151 |
| Membership * | −0.048 | 0.191 | −0.25 | 0.801 | −0.422 | 0.326 | 0.705 |
| PctOnfarmIncome | −0.001 | 0.003 | −0.37 | 0.713 | −0.006 | 0.004 | 70.711 |
| Telephone * | 0.544 | 0.354 | 1.54 | 0.124 | −0.150 | 1.238 | 0.936 |
| Distance | −0.011 | 0.008 | −1.36 | 0.174 | −0.028 | 0.005 | 4.279 |
| FarmArea | 0.351 | 0.134 | 2.63 | 0.009 | 0.089 | 0.613 | 0.93 |
| EducHoH | 0.005 | 0.015 | 0.29 | 0.774 | −0.028 | 0.037 | 6.932 |
| iRainfed* | 0.112 | 0.183 | 0.61 | 0.54 | −0.247 | 0.471 | 0.63 |
| Weatherinfo* | −0.291 | 0.175 | −1.66 | 0.097 | −0.635 | 0.053 | 0.649 |

(*) dy/dx is for discrete change of dummy variable from 0 to 1.

**Table 6.** Marginal effect after multivariate probit analysis due to change in frequency of drought.

| Variable | dy/dx | Std. Err. | Z | P > z | 95% | C.I. | X |
|---|---|---|---|---|---|---|---|
| **y = Linear Prediction (Predict) = −0.692** | | | | | | | |
| AgeHoH | 0.015 | 0.01 | 1.52 | 0.129 | −0.004 | 0.033 | 50.624 |
| GenderHoH * | −0.081 | 0.408 | −0.2 | 0.842 | −0.880 | 0.718 | 0.069 |
| EducHoH | 0.007 | 0.018 | 0.4 | 0.689 | −0.029 | 0.044 | 7.074 |
| Tenure * | 0.539 | 0.312 | 1.73 | 0.084 | −0.072 | 1.149 | 0.868 |
| BorrowedYN * | 0.151 | 0.21 | 0.72 | 0.472 | −0.261 | 0.562 | 0.429 |
| NumAdultMale | −0.196 | 0.096 | −2.04 | 0.041 | −0.384 | −0.008 | 2.148 |
| PctOnfarmIcome | −0.002 | 0.003 | −0.6 | 0.548 | −0.008 | 0.005 | 69.606 |
| Telephone * | 0.854 | 0.469 | 1.82 | 0.069 | −0.065 | 1.773 | 0.931 |
| iRainfed * | 0.227 | 0.238 | 0.95 | 0.34 | −0.239 | 0.692 | 0.667 |
| Distance | 0.002 | 0.013 | 0.17 | 0.865 | −0.024 | 0.028 | 4.352 |
| FarmArea | 0.464 | 0.129 | 3.59 | 0 | 0.211 | 0.718 | 0.929 |

(*) dy/dx is for discrete change of dummy variable from 0 to 1.

## 4. Discussion

### 4.1. Household Characteristics

Age of HoH was mostly found to be positive for all choices against all changes, meaning that the older the head of the household, the more likely the chance for adaptation. However, the negative coefficient of age for various adaptations might be supported by the fact that youth are relatively innovative compared to older generations [13,37], and the positive association of age and adaptation practices might be due to experience, cumulative knowledge and skills [38]. Furthermore, as noted in the literature, the influence of age was both positive [16,17,38] and negative [13,18,37,39].

Gender of HoH was commonly found to be negatively associated for all adaptation choices except for change in crop date adjustment against change in temperature and precipitation. This signifies that female-headed households were less likely to adopt the given choices. Nevertheless, a few studies have noted that female-headed household were expected to adopt crop diversification due to greater experience and more knowledge of various management practices [18]. Likewise, male-headed

households were highly adaptive to resource-intensive decisions, like changes in crop type and irrigation investment due to resource endowment [16]; this was probably also a consequence of social structure.

Except for the crop type change against temperature and precipitation alteration, education of HoH was positively associated with all adaptation options against all changes. In general, the higher the education, the higher the chance of adaptation. The majority of previous literature also reported mixed results for education [13,17,38].

In the temperature model, household size was negatively associated, except for change in crop variety, although this was only significant for change in crop type. This implies that a large household reduces the likelihood of changing crop type. Several previous studies also produced mixed result on the influence of household size [18,19,40].

Percentage of total income coming from the farm was positively correlated with all adaptation choices except change in crop date. However, in the drought model, for all choices, this was not significant. More importantly, this means that the higher the income coming from the farm, the greater the likelihood of adoption of adaptation alternatives. Apparently, the higher the percentage of farm income, the lower the likelihood of diversification; consequently, a higher dependency on farming leads to higher chances of adaptation [16].

The farm decision was negatively associated with all coping strategies except crop date adjustment, and was significant for irrigation investment. This suggests that HoHs were not interested in a change in farming system. As previously discussed in the age factor, the older generation is reluctant to embrace change. Moreover, education level might be a factor; the older generation is generally less educated, and those who are educated are more likely to adopt adaptations.

## 4.2. Farm Characteristics

In all cases, adaptation choices and farming area were positively associated, implying increasing area elevates the chances of adaptation. Similar results were reported by some previous studies [35]. However, several other studies reported mixed results [13,18,38].

For the temperature model, the absence of additional plots reduced the chance of adaptation, except for change in crop type. However, only irrigation investment was significant. If the household had additional plots, they were supposed to diversify the crop, which discouraged them from investing in irrigation, which might be associated with the cost of implementing irrigation on fragmented land. Various studies also noted a mixed influence for additional plots [18].

For the exclusively rain-fed farming system, all adaptation options were positively associated except for a few, which were not significant. This signifies that exclusively rain-fed farming is more vulnerable, encouraging farmers to adopt adaptation options.

In the case of tenure, generally, if farmers were farming on their own land, they were found to be more adaptive, except for change in crop type against precipitation alteration. This implies that farmers hesitate to invest in others' land, despite climatic adversities. Similar results were reported previously [19,41].

The number of adult males on the farm was positively associated in the temperature model, negatively associated in the drought model and not significant in the precipitation model. Also in previous studies, the influence of male adults working on farms was not consistent.

## 4.3. Household Access

The credit access of the household was found to be positive for all choices in all models, except change in crop type against temperature alteration. However, this was not significant. Thus, it implies that having credit access increased the likelihood of adaptation. A few previous studies reported all positive or mostly positive association of credit access in adaptation [17–19,38].

Having telephone access to the household was found to be positively associated with all adaptation options. A telephone is two-way communication, and more effective than one-way communication.

The positive relationship might be the result of the interaction between famers and agro-vets, which sell inputs in their locality.

The distance to the nearest market was generally negatively correlated to all adaptation options except for a few, which were not significant. Ordinarily, the relationship is the farther the markets, the lesser the likelihood of adaptation, because of the lower level of commercialization. However, the positive influence of market on change in crop type could be associated with crop diversification [18].

Membership was defined as any kind of affiliation of the household in the community, like cooperatives and farmers groups, which are, generally, positively associated with all adaptation options. Normally, membership increases the exposure of the household, which may encourage adaptation. Nonetheless, several previous studies produced similar mixed results for memberships [13,16,18,37].

Surprisingly, access to the weather information was negative in all cases, which could be due to ineffective weather information systems and a lower dependency of farmers on provided weather information, or ignorance by famers due to lower reliability. However, previous studies also noted a negative relationship of weather information on certain adaptation options [13,16,18,38].

### 4.4. Climate Perception Factors

The perception of temperature change was used in the precipitation model only; it produced mixed results. Its presence was expected to increase adoption of crop date adjustment and crop type alteration, but also to indicate unlikelihood of adopting change in crop variety and irrigation investment.

When farmers perceive change in drought frequency, they are expected to adopt given sets of coping strategies.

All adaptation options were negatively linked with a perceived decreased in precipitation; except for change in crop type, all were highly significant, which was probably connected to limited irrigation facilities and higher dependency on rainfall. Consequently, limited irrigation facilities and decreasing rainfall were expected to encourage farmers to invest in irrigation.

### 5. Conclusions

The perception of climate change was assessed, the factors influencing climate change adaptation were identified and its marginal effect was determined using the multivariate probit model for Nepal. The large majority of respondents replied that they had observed alterations in climatic variables—temperature, precipitation and drought. However, the direction of these changes was not univocal. Among others, gender, age and education level of the HoH, credit access, exclusively rain-fed farming, farm area, weather information, and additional plots were found to influence perception. Furthermore, the influence of the gender of the HoH on perception was not uniform. Despite differences in proportions, a greater number of the younger generation stated that they had observed changes. Regarding educational influence, generally, the higher the education, the higher the response for conforming changes. Similarly, experience, a composite of education and age, was positively associated with perception. Despite wider perception of climate change, adaptation was not common, i.e., less than a 50% adoption rate. Change in crop variety was the most popular measure against change in temperature; however, change in planting date was common against change in precipitation. As expected, irrigation investment was found to be the best adaptation option regarding the increased frequency of drought.

The likelihood ratio test of independence of error terms was rejected in all four MVPs, which justified the choice of the MVP over a single equation-probit model to capture the interaction of the adaptation options. For all MVPs, the four given adaptation options were complementary to each other, since pair-wise correlation coefficients are positive.

For the superior adaptability, focusing on crucial factors is extremely essential. Age, gender and education of HoH, credit access, farming area, tenure and irrigation are major determinants for all models and adaptation options. Therefore, targeting experienced farmers for the promotion

of adaptation options increases the likelihood of adoption. Credit access also plays a crucial role in increasing the likelihood of adaptation. This could be coupled with membership, which might increase the likelihood of adaptation. The strong positive association of farming area and adaptation options indicated that land consolidation was likely to favor climate change adaptation. Conversely, land fragmentation discouraged farmers from adopting comping measures. The results also identified that exclusively rain-fed and a higher percentage of farm income in total household income favored adaptation, indicating that poor farmers and those with no irrigation facilities were suffering severe consequences of climate change. Therefore, the government is encouraged to support poor farmers by offering year-round reliable irrigation facilities.

**Author Contributions:** A.G. conceptualized, investigated, wrote the original draft, did the review and editing, visualized and ran software for analysis. J.-H.Y. supervised and validated along with methodological work. All authors have read and agreed to the published version of the manuscript.

**Funding:** This research received no external funding.

**Acknowledgments:** We would like to acknowledge the Economics of Climate Change Adaptation (ECCA) Project of the UNDA for the data availability.

**Conflicts of Interest:** The authors declare no conflict of interest.

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
