# Peer review of "Perception to Adaptation of Climate Change in Nepal: An Empirical Analysis Using Multivariate Probit Model"

_sci, doi:10.3390/sci2040087_

Round 1

Reviewer 1 Report

Quality of introduction should be improved, for example the relationship between climate change, SDGs, and agriculture is not clearly, just focus on the topic “Climate change is affecting agricultural production”. Literatures review need further improvement, not just “measuring determinants are still relatively scarce [9]. Among them, the majority of studies are focused in Africa [11–15]”. For example, China has done a lot of research on climate change and agriculture, including measuring determinants. https://www.mdpi.com/2071-1050/11/7/1928/pdf Introduction section does not lead to scientific problems very well, and the literature review needs to be strengthened. Detailed description of the process of data collection and sampling should be given. For example, how to ensure random sampling. Description of Fig 1 have some mistakes, please correct. Models description should be added, explain why some variables were controlled, while some were not in other models. Further comparative analysis is needed, including the influencing factors on behaviors among models, and marginal effects.

Author Response

Quality of introduction should be improved, for example the relationship between climate change, SDGs, and agriculture is not clearly, just focus on the topic “Climate change is affecting agricultural production”. Literatures review need further improvement, not just “measuring determinants are still relatively scarce [9]. Among them, the majority of studies are focused in Africa [11–15]”. For example, China has done a lot of research on climate change and agriculture, including measuring determinants. https://www.mdpi.com/2071-1050/11/7/1928/pdf Introduction section does not lead to scientific problems very well, and the literature review needs to be strengthened. Detailed description of the process of data collection and sampling should be given. For example, how to ensure random sampling. Description of Fig 1 have some mistakes, please correct. Models description should be added, explain why some variables were controlled, while some were not in other models. Further comparative analysis is needed, including the influencing factors on behaviors among models, and marginal effects. Dear Reviewer Warm Greetings! Thank you very much for your all supports and wonderful review. As you have suggested on your reviewer's comment, We have updated our manuscripts. Specifically, introduction and literature review have been updated. Thank you

Reviewer 2 Report

Despite the fact that the manuscript is relevant the results must be supported by graphs. These missing information will clarify various conclusions of the paper.

Author Response

Despite the fact that the manuscript is relevant the results must be supported by graphs. These missing information will clarify various conclusions of the paper. Dear Reviewer Warm Greetings We really appreciate your contribution to make our paper much improved. We have tried to improved our manuscript inline with your suggestion. However, we would be glad if we could find any appropriate graphs to insert in results section. Thank you Best Regards

Reviewer 3 Report

This is an interesting article. The only real issue is that there are still some improvements in the English that would be useful. But the article is well-structured and the results interesting and useful.

There are many research projects that have investigated agricultural adaptatron to climate change and variability, and it would certainly be useful (not in this article) to make some comparisons between these different research processes.

Farmers' cultures can vary between different countries and even between territories in the same country. We often end up stressing the importance of human values (especially the human values of farmers) and depending on the culture on how farmers can sometimes develop their own strategies to investigate adaptation strategies and what can be the most useful adaptation sttrategies and the best way (for them) to determine the most productive adaptation strategies and the best sources of information about climate change and variability. 

Round 2

Reviewer 1 Report

After revision, the quality of manuscript have been significantly improved. To this end, I recommend that it be published in this journal.